# Emergency Preparedness for the COVID-19 Pandemic: Social Determinants Predicting the Community Pharmacists’ Preparedness and Perceived Response in Malaysia

**DOI:** 10.3390/ijerph19148762

**Published:** 2022-07-19

**Authors:** Tan Yu Xin, Kingston Rajiah, Mari Kannan Maharajan

**Affiliations:** 1Master in Pharmacy Practice, School of Postgraduate Studies, International Medical University, Kuala Lumpur 57000, Malaysia; 00000019033@student.imu.edu.my; 2GITAM School of Pharmacy, GITAM Deemed University, Hyderabad 502329, India; 3Department of Pharmacy Practice, School of Pharmacy, International Medical University, Kuala Lumpur 57000, Malaysia; 4School of Pharmacy, University of Nottingham Malaysia, Selangor 43500, Malaysia

**Keywords:** pharmacy workforce, community pharmacist, coronavirus, pandemic

## Abstract

Background: Pandemic preparedness of healthcare providers helps to mitigate future threats such as spread and fatality rates, as well as the management of the disease. Pharmacists are key partners with public health agencies, and the role of community pharmacists is becoming increasingly recognised in this COVID-19 pandemic. The study aimed to explore the emergency preparedness of community pharmacists (CPs) for COVID-19. Methods: A cross-sectional study was performed among community pharmacists using cluster sampling followed by convenient sampling. A self-administered questionnaire was formulated using references from the previous literature and the WHO preparedness checklist. Descriptive analysis was undertaken for the participants’ socio-demographic characteristics. All the data collected were entered into the Statistical Package for Social Sciences version 24 (SPSS V.24), (IBM SPSS Statistics for Windows, Version 24.0. Armonk, NY: IBM Corp.) for analysis. Results: Most of the CPs had five or fewer years of practice experience, and they had all the mandatory information relating to the needs of their communities regarding the disease. The participants knew where to acquire these resources whenever needed. They were able to recognise the signs and symptoms of the disease. Most participants felt that they were confident to provide patient education and carry out their duties during these challenging times. There was a strong position correlation between preparedness and the perceived response of the participants. Conclusion: The community pharmacists in Malaysia are prepared enough for COVID-19 pandemic management and perceive that they can respond during any unprecedented situations, such as COVID-19. Community pharmacists were aware of the challenges that they need to face in their community regarding COVID-19.

## 1. Introduction

The COVID-19 pandemic has placed enormous strain on the healthcare systems globally and exposed long-standing gaps in public health emergency preparedness [1]. An overburdened health system during the COVID-19 pandemic faces multiple challenges in responding to the consequences related to the health and safety of the public, as well as of health professionals [2]. A well-trained healthcare provider is essential for the health safety of a community, and their preparedness helps to mitigate threats, such as disease spread and fatality rates in the future. An effective, well-designed, coordinated multi-agency response across various organizations (Centers for Disease Control and Prevention (CDC), World Health Organization (WHO), etc.) is critical when facing any pandemic situation [3,4,5]. To enhance the preparedness of the healthcare systems and healthcare practitioners to deal with any public health emergency [6], organizations must be prepared to provide patient-centred and public-focused care. An improvement in training is considered as an actionable recommendation to improve the preparedness of front-line healthcare professionals.

Pharmacists are one of the key partners with public health emergency and responses, including vaccination, pharmaceutical care, health promotion and medication safety during pandemics. Their level of training and knowledge on medications/therapies and emergency preparedness is very useful in the COVID-19 response. They displayed a high level of willingness to participate in emergency training and assist in case of emergencies [7]. Pharmacists are also the most accessible and essential health care providers [8,9].

With the increased burden on primary care providers and hospital resources to manage COVID-19 patients, the role of community pharmacists (CPs) is becoming increasingly recognised during the COVID-19 pandemic. In most of the community care practice, CPs have now become the first point of contact [10,11,12] and especially in the management of minor ailments and health services, a high level of patient satisfaction has been achieved by the CPs [13,14]. Most CPs have established good relationships with their respective communities over time; they are, therefore, uniquely positioned to assist government bodies in the pandemic response at the community level. The duties and responsibilities of CPs regarding medical and non-medical products have been clearly outlined by the International Pharmaceutical Federation (FIP). In addition, they are held responsible for educating the public on prevention measures as well as referrals for suspected cases [15].

The engagement of community pharmacists in the four key disaster management phases—prevention, preparedness, response, and recovery (PPRR)—has drawn significant interest in recent literature owing to their unique positioning in the community [16]. The significant roles played by CPs during the COVID-19 pandemic in China and their health emergency preparedness and response efforts against the COVID-19 pandemic have been acknowledged [17]. A Delphi study defined the range of roles that pharmacists could play throughout these four phases [18]. Cadogan and Hughes have stressed the need to review and recognise the expertise among CPs as a unique member of total healthcare delivery [10]. Interestingly, McCourt and Mallhi have reported that not only is there limited literature on pharmacists’ preparedness but also the extent of their preparedness related to their practice [19,20].

Malaysia has been hit by multiple waves of COVID-19, and the country contained the COVID-19 outbreak during the first phase very well, owing to the prompt public health actions implemented by the Ministry of Health Malaysia and the Malaysian government. The Global Health Security Index 2019 ranked Malaysia 3rd in Asia in terms of the overall readiness to face a disease outbreak. However, Malaysia marked the highest daily toll, with 9020 cases reported on 29 May 2021, since the onset of the pandemic [14]. To contain the epidemic trajectory, the government has announced various measures, such as movement control order (MCO), in phases across the country. The National COVID-19 Immunisation Programme was rolled out on 24 February 2021, by prioritizing the healthcare professionals (HCPs) and frontliners, followed by the vulnerable populations. Along with other healthcare professionals, pharmacists are also involved in various healthcare services for the management of COVID-19. They play a vital role in closing the gaps that are exacerbated by the additional strain on the healthcare system.

Given the considerably significant scope of CPs in the COVID-19 pandemic, it is thus imperative to study the operational readiness capacities of CPs to aid the frontline national response to COVID-19. In contrast to HCPs who are based in hospital settings, where their associated risks are well-anticipated and prepared for [21], the risk to CPs and how well-prepared they are in overcoming an emergent situation are still poorly understood. Hence, this study explored the emergency preparedness of CPs for COVID-19 to inform health policy makers’ future decisions about modulating the COVID-19 pandemic management strategies in community pharmacies, as well as providing mandatory interventions to strengthen the preparedness capacities of CPs in halting the spread of COVID-19 and future public health threats.

## 2. Materials and Methods

### 2.1. Informed Consent and Ethical Approval

The study was approved by the International Medical University Joint Committee for Research and Ethics MPP 1/2020 (10). Participation in the study was voluntary and written consent was obtained from all the participants. Anonymity and confidentiality of the participants were guaranteed. Only one pharmacist was included from each community pharmacy. Pharmacists unable to give consent for any reason were excluded from this study.

### 2.2. Study Design

This cross-sectional study was conducted between December 2020 and March 2021, in Kuala Lumpur and Selangor, Malaysia, where a higher number of cases were reported.

### 2.3. Sample Size

The study sample size was calculated using an online Raosoft calculator. A minimum of 149 participants was required with an assumption of a 5% margin of error and a 95% confidence interval. A total of 152 CPs completed the questionnaire, which met the required sample size.

### 2.4. Sampling

Cluster sampling followed by convenient sampling was undertaken [19]. Geographic segmentation was based on districts and regions and used as clusters. In the first stage, the district clusters were selected by a simple random method. In the second stage, from each selected district, regions were selected by a simple random method. In the third stage, from each selected region, the community pharmacies were approached conveniently to participate in the study.

### 2.5. Study Questionnaire

A self-administered questionnaire was formulated using references from the previous literature and WHO preparedness checklist [22,23]. It was designed in the English language as it is the medium of learning for the pharmacy profession in Malaysia.

The questionnaire consisted of 35 items, which were grouped into three different sections. Section A contained nine questions to gather the respondents’ sociodemographic data; Sections B and C comprised 15 and 11 close-ended questions, respectively, to measure their preparedness and perceived response towards the COVID-19 pandemic. Items in Sections B and C were answered on a ‘Yes/No’ basis. One point was given for the “Yes” response and zero for the “No” response. The responses were then summed for a total score ranging from 0–15 for Section B and 0–11 for Section C. The cut-off point was 50% [6] to define the CPs’ preparedness and response levels. Participants who scored ≥8 on the preparedness scale were considered as “adequately prepared”, while those who scored <8 were considered as “inadequately prepared”. Likewise, a score ≥6 on the response scale indicated the pharmacists were able to respond adequately, otherwise categorised as the pharmacists unable to respond adequately.

### 2.6. Validity and Reliability of the Study Questionnaire

The questionnaire was validated by three subject experts and revisions were made based on the comments prior to implementation. Among the three experts, one of them is a Malaysian community pharmacist with 25 years of experience. Another expert has 20 years of experience in social pharmacy and behavioural sciences research. Another expert is an associate professor in pharmacy practice with 18 years of experience. Revisions were made to the questionnaire based on the comments given to ensure its appropriateness. The reliability was determined by piloting among 20 CPs, and the Cronbach’s alpha coefficient was 0.72 and 0.76 for preparedness and perceived response, respectively. The overall Cronbach’s alpha of 0.74 indicated an acceptable internal consistency. The study questionnaire has been included as a Appendix A.

### 2.7. Data Analysis

Descriptive analysis was undertaken for the participants’ socio-demographic characteristics. All the data collected were entered into the Statistical Package for Social Sciences version 24 (SPSS V.24) (IBM SPSS Statistics for Windows, Version 24.0. Armonk, NY: IBM Corp) for analyses. Descriptive analysis was applied to describe the respondents’ socio-demographic profile by presenting the categorical variables in frequencies and percentages. The direction and strength of association among preparedness and response (PR) scores and between PR scores with socio-demographic variables were tested via rank biserial correlation and Spearman rank with the significance level set at *p* < 0.05. Socio-demographic factors that were considered in this study included age, gender, ethnicity, state, years of professional experience, type of pharmacy, number of pharmacists in the pharmacy, disease management training, and the number of COVID-19 cases attended. We used the rank biserial correlation to find the relationship between gender, ethnicity, type of pharmacy, state, (categorical variables), and PR scores (interval/ratio). We used Spearman’s rank to find the relationship between age, experience in years, number of pharmacists in a pharmacy, state, trained on disease outbreak management, number of COVID-19 cases attended (continuous variables), and preparedness and PR scores (interval/ratio).

## 3. Results

Table 1 shows the socio-demographic profile of the participants. Overall, the participants were predominantly female (65.8%). Most of the participants belonged to the age group of 21 to 30 years (61.2%) and most of them were Chinese (66.4%). In terms of professional experience, most of the CPs had five or fewer years of experience in practice (58.6%). Many of the CPs were from chain pharmacies (73.0%). More than half of the participants (55.3%) had never received any training on disease outbreak management before. Very few participants (13.2%) had dealt with COVID-19 cases more than once, and 6.6% of them have dealt with at least one COVID-19 case.

Table 2 represents the participants’ preparedness towards COVID-19. All participants agreed that clients’ recent travel or residence history should be collected, given the potential exposure to COVID-19 from any COVID-19 hotspots, with the highest mean score (1.00 ± 0.000). Favourably, a large proportion of participants were conscious of the challenges they need to conquer from their surrounding communities regarding COVID-19, with a higher mean score (0.96 ± 0.195). Most of the participants reported that they had all the mandatory information related to the needs of their communities regarding the disease (0.91 ± 0.28), and they knew where to acquire these resources whenever needed (0.87 ± 0.339). Additionally, the participants knew the emergency contact of the local authority (0.80 ± 0.399) and how to use personal protective equipment (PPE) (0.89 ± 0.316) as well as to isolate the suspected person (0.76 ± 0.427) in case of an emergency situation at their premises. On the other hand, very few participants have recently participated in educational activities related to COVID-19 preparedness, with the lowest mean score (0.44 ± 0.498). A relatively low number of participants reviewed COVID-19 related literature (0.53 ± 0.501). The overall preparedness, as measured by the total mean score, was 10.58 ± 0.384, which was more than the 50% threshold.

Table 3 represents the participants’ perceived responses to COVID-19. The data revealed that most of the participants were able to recognise the signs and symptoms of the disease, with the highest mean score obtained (0.95 ± 0.224). Most participants felt that they were confident in providing patient education (0.90 ± 0.299) and carrying out their duties (0.85 ± 0.360) during these challenging times. On the other hand, only a few participants felt that they could care for infected patients independently without direct supervision (0.36 ± 0.480) or respond to patients who developed worsening symptoms or reactions of COVID-19 (0.23 ± 0.422). A relatively low mean score (0.41 ± 0.494) was obtained when the participants were questioned on their ability to detect potential COVID-19 clusters evidenced by mass exposure with similar symptoms. Fewer than half of the participants felt that the availability of therapies on their premises was sufficient to manage COVID-19 emergencies (0.46 ± 0.500). The overall perceived response, as measured by the total mean score, was 6.28 ± 0.469, which was just above the 50% threshold.

Table 4 shows the results of correlation analyses. There was a strong position correlation between preparedness and the perceived response of the participants (r = 0.502, *p* = 0.002). Among the socio-demographic variables, there was a strong positive correlation between participants who were trained in disease outbreak management and their preparedness (r = 0.542, *p* = 0.001). There was a moderate positive correlation between participants who were trained in disease outbreak management and their perceived response (r = 0.366, *p* = 0.03). There was a weak positive correlation between the participants with long working experience and their preparedness (r = 0.295, *p* = 0.013). There was a weak positive correlation between the age of the participants and their perceived response (r = 0.216, *p* = 0.03) There was a weak positive correlation between participants who had attended COVID-19 cases and their perceived response (r = 0.271, *p* = 0.031). Other variables, such as gender, ethnicity, type of pharmacy, state, and the number of pharmacists, did not significantly correlate with the participants’ preparedness and response.

## 4. Discussion

The COVID-19 pandemic has rekindled attention to the level of emergency preparedness and readiness among HCPs globally. Owing to the proximity to the public, CPs are ideally positioned to respond to the COVID-19 pandemic through their multifaceted roles in the healthcare system. Meanwhile, their risks of encountering an infected patient and further provoking the infection chain are also probable at the workplace, as they must interact face-to-face with the public in the provision of pharmaceutical services. Therefore, CPs’ strategic preparedness and response to counter the COVID-19 pandemic should be ensured to preserve the safety of the local community.

Most of the participants were female, young, and had five years or less of work experience, suggesting that many would not have prior experience in responding to previous infectious outbreaks, such as SARS-CoV (2003) and H1N1 influenza (2009) [24]. The study demographic also reflected the cultural diversity in the workforce of Malaysia, of which pharmacists belonging to Chinese ethnicity made up the most. Ong et al. (2018) also reported similar results, denoting that the present demographic distribution was practically illustrative of the CP population in the selected study regions [25]. The results reflected the fact that very few pharmacists are trained in disease outbreak management. Hence, the health authorities in the country should be looking forward to providing training for pharmacists so that they will be better prepared to face future pandemics.

Most of the participants perceived that they could respond effectively to the ongoing pandemic, and our results are in line with a previous report by Kua and Lee (2021) [24]. This affirms that pharmacists could be instrumental during this public health crisis. The study was conducted at a later phase of the COVID-19 pandemic, in which pharmacists have already initiated the necessary preparedness actions. The preparedness level of pharmacists for COVID-19 was higher than that reported previously for Zika infection in Malaysia [6]. This can be ascribed to the alarming rates of COVID-19 transmission and its continued global death count, compelling HCPs, including CPs, to take this pandemic seriously. Gowing et al., (2017) reported that preparedness and response may vary among HCPs based on the situation [26].

The participants reported that they had all the required COVID-19 related information. However, the questionnaire used in this study did not examine their primary sources of information. A study by Kara et al. (2020) revealed that the information sources utilised by the pharmacists to learn about COVID-19 infection influenced their knowledge and attitudes towards the disease, which consequently affected their ability to convey factually accurate information to the public [27]. The exponential rise in demand for and dissemination of information regarding COVID-19 has resulted in the “infodemic” phenomenon, which is highly prone to misinformation and falsehoods [28]. Hence, CPs should carefully evaluate their source of information to ensure the authenticity and reliability. Notably, the present findings showed that only some of the participants read journal articles on COVID-19-related topics, which is a concern as journal articles are among the trustable sources of quality information. This agrees with the findings of ElGeed et al., (2021) [29]. Given that almost all COVID-19 publications are freely accessible nowadays, the low reading rate and poor participation in educational activities could be due to the lack of time, workload, and the shortage of the pharmacy workforce during these unprecedented circumstances.

CPs reported that there is a need to take patients’ travel or residence history, as this could help in identifying potentially high-risk patients for COVID-19. However, ElGeed et al., (2021) reported that, in real practice, only a few CPs gather history from their patients to assess their health status; the authors justified this by stating that their scope of practice was still limited to traditional approaches with little consideration given to patient-centred care [29]. The role of CPs is now expected to be more patient-oriented, shifting beyond the traditional product-centred practice of the mere dispensing of medicines. Moreover, it is encouraging that CPs perceived that they could identify the common signs and symptoms of COVID-19 and were sufficiently prepared to isolate the suspected persons on their premises. Similar results were observed in other studies [30,31]. This could help in the early identification of COVID-19 patients and avoid delays in treatment, which could lead to severe complications.

The emergence of COVID-19 has caused an excessive demand for PPE, such as face masks, gloves, and goggles as well as other medical supplies, and there was a global shortage during the early containment phase of the pandemic. Many pharmacies were confronted with challenges in restocking these products; this was exacerbated by the surge in panic buying, where consumers buy unusually large amounts of a product in anticipation of, or after, a disaster or perceived disaster. There was also an observed tendency of change in the rational use of medications among the population due to their false beliefs about prescription-only medicines as a cure for COVID-19 [32]. In the present study, it was reassuring to note that the CPs reported that they had adequate supplies of PPE to manage the emergencies and carry out their duties. Moreover, the CPs had the necessary information regarding the situation and knew whom to contact in order to acquire the resources. All of these could be the outcome of the guidelines that outlined the steps for pandemic preparedness [33]. However, the pharmacists found it challenging to independently recognise cases independently, recognise clusters, and respond to those with severe symptoms. Moreover, the CPs were unable to detect potential COVID-19 clusters. As most of them were early career CPs with minimal years of practice, they may perceive themselves to be less competent and lack confidence in the decision-making process. Some reported that this could have been the result of minimal years of practice, new practitioners, being less competent, and lack of confidence in the decision-making process. In addition, this study does not have the information on which type of medication CPs are entrusted with, concerning the appropriate dispensing and drug procurement in the right quantities to guarantee the supply round the clock. Vigilant observation of any unusual selling trends due to self-medication, behaviour, or hoarding of medicines by the public could prevent drug shortages that may negatively impact patient care. Badreldin et al., (2021) suggested that drug demand analysis should be conducted in a parallel manner to identify the medications of interest in the context of the pandemic, in order to ensure the availability of these essential medicines [34]. Another study stated that CPs are in a privileged position to provide timely and real-world data on medicine sales and shortages; this could help the national authorities to take coordinated actions to sustain the integrity of the national market [22].

Disease outbreak management training was found to have a primary influence on the CPs’ preparedness and perceived response to the COVID-19 pandemic. This relationship had been reported by earlier studies as well [35,36]. The disease outbreak preparedness interventions in the form of education, drills, or training can improve the knowledge, confidence, and workforce collaboration in connection with the actual disasters, which may be translated into a high preparedness level. Thakre et al., (2020) affirmed that training is an ideal way of imparting knowledge and creating awareness among HCPs [37]. However, more than half of the CPs in this study had not previously been involved in any form of training related to disease outbreak. It reflects the current deficiencies in the planning, publicity, and governance of these initiatives in the private sector, which calls for the attention of the health policymakers and related stakeholders to investigate this at the earliest opportunity.

Consistent with prior research [24], the age and the years of professional experience of the respondents were other factors that correlated with their perceived response to the pandemic. This could be explained by senior members of the profession being more likely to have experiential learning over time; this would prepare them for better resilience and response to emergencies; moreover, the decision-making process is better among experienced pharmacists [38]. This also relates to the correlation between the number of COVID-19 cases attended by the CPs and their perceived response attributes. CPs who had encountered COVID-19 patients may have a higher risk perception. This could have improved their precautions and responsibilities to prepare sufficiently for their next encounter with another COVID-19 patient.

In line with previous studies [39,40], the relationship between gender and emergency preparedness was found to be insignificant. This could be explained by the equality between males and females in terms of their decision-making rights, access to resources, and participation in educational activities. However, the result contradicts some studies which stated that male respondents had higher COVID-19 knowledge scores and were generally more prepared during pandemics in comparison to female counterparts [41]. It has also been suggested that there would be some disparities among individual preparedness with respect to pharmacists’ races; this might be related to other factors that correlate with preparedness, such as income level, religious values, and cultural differences [42]. Nonetheless, there is no significant influence of the ethnicity on the CPs’ emergency preparedness.

### Strengths and Limitations

Our study revealed that pharmacists can be an integral part of the management plan in a pandemic, due to their extensive knowledge and experience. Many pharmacists may have been trained to handle such situations. In this study, the self-administered questionnaire validated their desire to effectively engage in the COVID-19 pandemic. As this study used dichotomous questions, the respondents did not get the opportunity to share their opinions. The samples were taken from two states in Malaysia and hence caution should be taken while generalizing the data. Convenience sampling was done at the end stage of cluster sampling. As it is a mix of probability and non-probability sampling methods, there are chances for selection bias at the end stage. The time duration was a resource limitation in this study as the preparedness and the response of the participants needed to be obtained within a limited time frame.

## 5. Conclusions

The community pharmacists in Malaysia are sufficiently prepared for any pandemic and perceive that they can respond to any unprecedented situations. They were aware of the challenges they face in their community during such pandemics. They could identify the signs and symptoms of COVID-19 and provide counselling. Pharmacists with years of experience, senior in age, those who have attended at least one COVID-19 patient, and undergone disease management training are prepared and can respond well to the pandemic situations. Further studies could be conducted to identify the problems faced by pharmacists during and after the pandemic.

## Figures and Tables

**Table 1 ijerph-19-08762-t001:** Socio-demographic characteristics of participants (*n* = 152).

Characteristic	Frequency (*n*)	Percentage (%)
**Gender**		
Male	52	34.2
Female	100	65.8
**Age**		
21–30	93	61.2
31–40	46	30.3
41–50	8	5.3
>50	5	3.3
**Ethnicity**		
Malay	42	27.6
Chinese	101	66.4
Indian	9	5.9
**Experience in years**		
≤5	89	58.6
6–10	45	29.6
>10	18	11.8
**Type of pharmacy**		
Chain	111	73
Independent	41	27
**State**		
Kuala Lumpur	74	48.7
Selangor	78	51.3
**No. of pharmacist in a pharmacy**		
1	26	17.1
>1	126	82.9
**Trained on disease outbreak management**		
Never	84	55.3
At least once	53	34.9
More than once	15	9.9
**No. of COVID-19 cases attended**		
None	122	80.3
1	10	6.6
>1	20	13.2

**Table 2 ijerph-19-08762-t002:** Participants’ preparedness scores towards COVID-19.

No.	Statement	Mean Scores ± SD
1	I have all the information related to the needs of my community regarding COVID-19.	0.91 ± 0.281
2	I am aware of the challenges that I need to face from my community regarding COVID-19.	0.96 ± 0.195
3	I know where to get the resources/materials needed for my community in this COVID-19 situation.	0.87 ± 0.339
4	I am aware of the programs regarding COVID-19 preparedness and management that are offered by the Ministry of Health.	0.68 ± 0.469
5	I read journal articles related to COVID-19 preparedness.	0.53 ± 0.501
6	I know whom to contact (chain of command) in a disastrous situation in my community.	0.80 ± 0.399
7	I find that the research information on COVID-19 management is easily accessible from my pharmacy setting.	0.64 ± 0.480
8	I have participated in educational activities dealing with COVID-19 preparedness recently (ex: continuing education, webinars, or conferences)	0.44 ± 0.498
9	I agree that history should be taken on whether the customers have resided in or travelled to a country.	1.00 ± 0.000
10	In case of emergency, I know how to use personal protective equipment.	0.89 ± 0.316
11	In case of emergency, I know how to execute decontamination procedures within the pharmacy.	0.78 ± 0.414
12	I am familiar with accepted triage principles used in emergency situations.	0.50 ± 0.502
13	In a case of emergency, I know how to perform isolation procedures to minimise the risks of community exposure.	0.76 ± 0.427
14	I consider myself prepared for the management of COVID-19 outbreak.	0.70 ± 0.461
15	I am ready for peer evaluation of my skills on preparedness to COVID-19.	0.63 ± 0.486
	Total mean score	10.58 ± 0.384

SD: Standard deviation.

**Table 3 ijerph-19-08762-t003:** Participants’ scores regarding their perceived response towards COVID-19 (*n* = 152).

No.	Statement	Mean Scores ± SD
1	I am confident in providing patient education on COVID-19.	0.90 ± 0.299
2	I can identify the signs and symptoms of COVID-19.	0.95 ± 0.224
3	I am confident that I can perform my duties in COVID-19.	0.85 ± 0.360
4	I can respond as a direct-care provider or first responder in COVID-19.	0.61 ± 0.490
5	I can manage COVID-19 patients independently without any supervision.	0.36 ± 0.480
6	I can respond to patients with worsened symptoms and reactions of COVID-19.	0.23 ± 0.422
7	There are enough medications needed to manage the COVID-19 emergency.	0.46 ± 0.500
8	There is enough PPE needed to manage the COVID-19 emergency.	0.53 ± 0.501
9	I personally have received clients needing help with COVID-19 issues.	0.45 ± 0.500
10	I can identify possible indicators of mass exposure evidenced by clustering of patients with similar symptoms	0.41 ± 0.494
11	I am ready for peer evaluation of my skills on responsiveness to COVID-19.	0.53 ± 0.501
	Total mean score	6.28 ± 0.469

SD: Standard deviation.

**Table 4 ijerph-19-08762-t004:** Correlation among PR and between PR and independent variables.

	Preparedness	Response
Response	**0.502 ****	-
**Independent variables**		
Gender	−0.051	−0.181
Age	0.142	**0.216 ***
Ethnicity	0.109	−0.019
Experience in years	**0.295 ***	0.033
Type of pharmacy	−0.008	0.058
State	0.195	0.100
Number of pharmacists	−0.089	0.006
Trained on disease outbreak management	**0.542 ***	**0.362 ***
Number of COVID-19 cases attended	0.151	**0.271 ***

** Correlation is significant at the 0.01 level (2-tailed). * Correlation is significant at the 0.05 level (2-tailed).

## Data Availability

The data presented in this study are available on request from the corresponding author.

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
