# Peer review of "Emergency Preparedness for the COVID-19 Pandemic: Social Determinants Predicting the Community Pharmacists’ Preparedness and Perceived Response in Malaysia"

_ijerph, 2022, doi:10.3390/ijerph19148762_

Round 1
Reviewer 1 Report
The manuscript entitled “Emergency preparedness for the COVID-19 pandemic: social determinants predicting the community pharmacists’ preparedness and perceived response” is well written and properly explained. Authors of this article have raised an important yet neglected sector in public health regarding pandemic and outbreak of diseases. Role of pharmacies and pharmacists are usually ignored in pandemic scenarios. Suggestions given regarding training of pharmacists to perform their important role in containing a pandemic are well explained and discussed in this manuscript. Language used in the article is easy to understand for all kinds of readers. Many important aspects regarding the preparedness and emergencies to be dealt by pharmacies are mentioned and discussed in detail.
My suggestions to improve the manuscript are:
- More details of the questionnaire should be given, which will help the readers to understand this article better.
- Suggestions should be incorporated regarding future studies. For example, a detailed survey should also be conducted to properly investigate the abilities of pharmacy staff to identify the patients of pandemic and further studies should be conducted to identify the problems faced by pharmacy staff during and after pandemic or outbreak.
Best wishes.
Reviewer 2 Report
Please see attached document.

Reviewer 3 Report
Dear Authors,
Thank you for the opportunity to review your manuscript focusing on pharmacists’ preparedness and perceived response related to the COVID-19 pandemic and exploring the social determinants. Though, the analysis presented here is timely since we are still grappling with COVID-19 and learning more about it every day, there are number of issues with the methodology and analyses, which certainly deserves authors’ attention. The major and minor comments on the manuscript are appended below.
1. Though, the focus of this research is to explore the Malaysian pharmacist emergency preparedness related to the COVID-19, the title and introduction is rather much broader and more generic in nature and not sufficiently and specifically in the light of engagement of the Malaysian community pharmacists within the Malaysian healthcare system thus creating an ambiguity about its title and research background.
2. Authors failed to describe the instrument that they have developed and used to measure emergency preparedness and perceived response in sufficient details rather than simply mentioning as created from the previous literature (self-citation only) and WHO preparedness checklist. This instrument development warrants more details to understand its comprehensiveness and scope for the benefit of its readers.
3. More frequent use of word disaster management (very broad term that encompasses different types of disasters – natural, man-made, and hybrid and so many other sub types), and emergency preparedness (here specifically addressing COVID-19) synonymously by authors creates further confusion about what authors were implying here in the manuscript in context of their objectives.
4. Intro- Line 37 – please provide the context here - looming high caseloads faces multiple challenges- you are referring to COVID-19 here, right?
5. Intro- Line 42- please replace word “health security” with health safety.
6. Authors should clarify about the discrepancy in their sample size calculation. How 5% margin of error corresponds with 90% CI%, when all the analyses were conducted using a priori significance level of <.05 with a 95%CI? Why authors chose 90%CI for calculation and not 95%CI?
7. Authors justification for sample size of 149 also raises further questions about what beta level, and effect size was considered? And why sample size calculation was not based on the number of questionnaire items where the sample size would have been much greater?
8. Authors did not provide any justification why dichotomous response scale of yes/no was used rather than Likert or any other type? And why the items were not grouped into domains rather than referring to single item as it appears they could have been easily grouped in to 2-3 domains for more robust analyses based on what they were assessing?
9. Line 133- please replace word “mark” with point(s).
10. Why did authors choose 50% cut-off point to define the CPs’ preparedness and response levels? The reference provided here is again a self-citation and looked at the emergency preparedness among healthcare providers (I am assuming here that it was among mixed healthcare professionals with different background, disciplines and experience) and not among the specific healthcare professional such as pharmacists (unique) for this research?
11. Line 144 – please provide specific academic background and experience of three subject experts who assessed the face and content validity of this questionnaire.
12. It is somewhat baffling to me, how authors were able to assess the correlation between gender, ethnicity, type of pharmacy, state, trained on disease outbreak management (categorical variables) and preparedness and PR scores (interval/ratio) using Spearman Rho correlation analysis (Table 4)?
13. Please align rows appropriately in Table 1.
14. How authors considered (r=0.506, line 203 and r=0.366, line 207) as a HIGH and MODERATE positive correlation? Is there any reference for such?
15. The limitation section needs further expansion - limited sample size, use of convenience sampling, resource limitations etc?
16. Authors broader conclusion that the CPs in Malaysia are sufficiently prepared for any infection pandemic and perceive that they can respond during any unprecedented situations like Covid-19, considering the 50% cut -off threshold value used (in the absence of any valid reference or justification) in this research appears to be far-fetched.
Round 2
Reviewer 3 Report
Dear Authors,
Thank you for addressing my comments and revising the manuscript.
I am partially satisfied with few of the authors’ revisions. However, there are still larger issues that are not addressed which made me to reject this current manuscript in first place. Here is some more critique considering the authors’ response to my comments, which supports my initial decision of rejection, which has not changed.
Authors’ response to Q. 2 is partially correct. The nature of COVID-19 pandemic is much different than the Ebola and Zika virus because of its etiology, nature of transmission and outreach of its spread. Considering COVID-19 pandemic, the questionnaire should have been much comprehensive and have been developed by including most recent published literature assessing pharmacists emergency preparedness for COVID-19, in particular, since this is the objective of this study. The WHO emergency preparedness checklist referred in the study was meant to assess the Ebola outbreak. Here are few citations assessing pharmacists’ emergency preparedness for COVID-19 that were published before this current manuscript in hand, and there are many more…..
Thong, K.S., Selvaratanam, M., Tan, C.P. et al. Pharmacy preparedness in handling COVID-19 pandemic: a sharing experience from a Malaysian tertiary hospital. J of Pharm Policy and Pract 14, 61 (2021). https://doi.org/10.1186/s40545-021-00343-6
Itani R, Karout S, Khojah HMJ, Jaffal F, Abbas F, Awad R, Karout L, Abu-Farha RK, Kassab MB, Mukattash TL. Community pharmacists' preparedness and responses to COVID-19 pandemic: A multinational study. Int J Clin Pract. 2021 Sep;75(9):e14421. doi: 10.1111/ijcp.14421. Epub 2021 Jun 8. PMID: 34053167; PMCID: PMC8236935.
Bahlol M, Dewey RS. Pandemic preparedness of community pharmacies for COVID-19. Res Social Adm Pharm. 2021 Jan;17(1):1888-1896. doi: 10.1016/j.sapharm.2020.05.009. Epub 2020 May 11. PMID: 32417070; PMCID: PMC7211678.
ASHP COVID-19 Pandemic Assessment tool- Available at: https://www.ashp.org/-/media/assets/pharmacy-practice/resource-centers/Coronavirus/docs/ASHP_COVID19_AssessmentTool.pdf
Authors’ response to Q. 7 is again partially correct. Sample size can be calculated considering various factors as stated by the authors. Moreover, sample size for survey research can be calculated based on probability (random) or controlled sampling (convenience/purposive sampling) requiring different parameters. In addition, the sample size can be calculated based on the number of questionnaire items and expected response rate too. For more info, please refer to this book.
Streiner DL, Norman GR. Health Measurement Scales. New York, NY: Oxford University Press;
2002.
Authors’ response to Q. 8 is not convincing enough. How authors determined if their items/scale was unidimensional measuring emergency preparedness and perceived response? On the face value, the items appears to be assessing multi dimensionality? Therefore one-dimensionality of the questionnaire is still remains in question?
Authors’ response to Q. 11 – Authors should include this info into the relevant section of the manuscript.
Authors’ response to Q. 12 is again inappropriate. The reference provided here came from an answer provided to the question asked on ResearchGate. Moreover, the referred answer came from a deleted profile. What is the authenticity of such reference and what about its accuracy? Is this reference even professionally acceptable?
Here is some more information on when to use Spearman’s Rho analysis.
Comparison of the Pearson and Spearman correlation methods
Available at: https://support.minitab.com/en-us/minitab-express/1/help-and-how-to/modeling-statistics/regression/supporting-topics/basics/a-comparison-of-the-pearson-and-spearman-correlation-methods/
Spearman's Rank-Order Correlation. Available at: https://statistics.laerd.com/statistical-guides/spearmans-rank-order-correlation-statistical-guide.php
Use of inappropriate statistical test for data analyses, further raises the question about the reliability and validity of the results presented by the authors for their correlation analyses which is a crux of this study. Even if this part is excluded from the manuscript, the current study will remain simply a descriptive study offering no benefits to its readers or add anything to the published literature.
Authors’ response to Q. 14 is partially true. The strength of the correlations and cut-off points are arbitrary and inconsistent (different source proposed different cut-off points) and should be used judiciously. Here are few references to expand on this----
Hinkle DE, Wiersma W, Jurs SG. Applied Statistics for the Behavioral Sciences. 5th ed. Boston: Houghton Mifflin; 2003.
Schober, Patrick MD, PhD, MMedStat; Boer, Christa PhD, MSc; Schwarte, Lothar A. MD, PhD, MBA Correlation Coefficients: Appropriate Use and Interpretation, Anesthesia & Analgesia: May 2018. Volume 126 - Issue 5 - p 1763-1768 doi: 10.1213/ANE.0000000000002864.
Authors’ response to Q. 16 is also partially correct. Since, the self-cited article authors referring to showed significant differences between different healthcare professionals (GPs, RPh, Nurses, dentist etc.) about the emergency preparedness and perceived response for Ebola outbreak. Pharmacists participated in this study had low mean score on both scale compared to general practioners and nurses. This again, doesn't adequately justifies the rationale for the use of 50% cut-off score.
Author Response
Response to reviewer 3 -Round 2
Comment: I am partially satisfied with few of the authors’ revisions. However, there are still larger issues that are not addressed which made me reject this current manuscript in first place. Here is some more critique considering the authors’ response to my comments, which supports my initial decision of rejection, which has not changed.
Response: The earlier response attempted to address the concern raised by all the reviewers. We improved our manuscript significantly and most of the concerns raised by the reviewer have been addressed now. Thanks for the critical review and the feedback on our manuscript. All the measures have been taken to address the concerns raised by the reviewer. We believe that this improvement met your expectation.
Comment: Authors’ response to Q. 2 is partially correct. The nature of COVID-19 pandemic is much different than the Ebola and Zika virus because of its etiology, nature of transmission and outreach of its spread. Considering COVID-19 pandemic, the questionnaire should have been much comprehensive and have been developed by including most recent published literature assessing pharmacists emergency preparedness for COVID-19, in particular, since this is the objective of this study. The WHO emergency preparedness checklist referred in the study was meant to assess the Ebola outbreak. Here are few citations assessing pharmacists’ emergency preparedness for COVID-19 that were published before this current manuscript in hand, and there are many more.
Response: Thanks for this extensive feedback. The WHO emergency preparedness checklist referred to in the study assessed the Covid 19 (not the Ebola outbreak). The study cited by the reviewer “Pandemic preparedness of community pharmacies for COVID-19” has also used the WHO checklist which is similar to this study.
Comment: Thong, K.S., Selvaratanam, M., Tan, C.P. et al. Pharmacy preparedness in handling COVID-19 pandemic: a sharing experience from a Malaysian tertiary hospital. J of Pharm Policy and Pract 14, 61 (2021). https://doi.org/10.1186/s40545-021-00343-6
Response: Thanks for this reference citation. This study did not use any questionnaire. It is a sharing experience by hospital pharmacists from Malaysia. Moreover, this study concludes that limited resources in the field of digital health and a lack of experience in disaster management warrant the attention of health policymakers to improve these aspects. Our study is focusing on disaster management during pandemics. Hence, the this study is very much valid as it focussed the COVID-19 pandemic preparedness from the Malaysian perspective.
Comment: Itani R, Karout S, Khojah HMJ, Jaffal F, Abbas F, Awad R, Karout L, Abu-Farha RK, Kassab MB, Mukattash TL. Community pharmacists' preparedness and responses to COVID-19 pandemic: A multinational study. Int J Clin Pract. 2021 Sep;75(9):e14421. doi: 10.1111/ijcp.14421. Epub 2021 Jun 8. PMID: 34053167; PMCID: PMC8236935.
Response: Thanks for this reference citation. This study used the questionnaire from the previous influenza vaccination (not specific for COVID-19). Moreover, this study also has similar questionnaire items just like our study. For example, Personal protective equipment, the experience of COVID‐19 symptoms, etc. As such, our study focused on community pharmacists' preparedness and responses to the COVID-19 pandemic in Malaysia.
Comment: Bahlol M, Dewey RS. Pandemic preparedness of community pharmacies for COVID-19. Res Social Adm Pharm. 2021 Jan;17(1):1888-1896. doi: 10.1016/j.sapharm.2020.05.009. Epub 2020 May 11. PMID: 32417070; PMCID: PMC7211678.
Response: Thanks for this reference citation. This study used the questionnaire from the public health guidelines by the WHO, International Pharmaceutical Federation (FIP), The pharmacy Guild of Australia, The British Columbia Pharmacy Association (BCPhA), Pharmaceutical Services Negotiating Committee (PSNC), and National Institute for Health and Care Excellence (NICE)). These guidelines are not specific for COVID-19 but for disaster management. Moreover, this study also has similar questionnaire items just like in our study. For example, the degree of symptoms reported by customers to community pharmacists, practice surrounding patient counselling etc. As such, our study focused on community pharmacists' preparedness and responses to the COVID-19 pandemic in Malaysia.
Comment: ASHP COVID-19 Pandemic Assessment tool- Available at: https://www.ashp.org/-/media/assets/pharmacy-practice/resource-centers/Coronavirus/docs/ASHP_COVID19_AssessmentTool.pdf
Response: This assessment tool is very much similar to the tool used in our study. For example, Integration with Institutional Planning, Departmental Leadership, Public and Professional Education and Training, Medications and Supplies, Staffing, and Public Affairs/Communications. As such, the questionnaire used in our study is valid and suitable for predicting the community pharmacists’ preparedness and perceived response to the COVID-19 pandemic in Malaysia.
Comment: Authors’ response to Q. 7 is again partially correct. Sample size can be calculated considering various factors as stated by the authors. Moreover, sample size for survey research can be calculated based on probability (random) or controlled sampling (convenience/purposive sampling) requiring different parameters. In addition, the sample size can be calculated based on the number of questionnaire items and expected response rate too. For more info, please refer to this book.
Streiner DL, Norman GR. Health Measurement Scales. New York, NY: Oxford University Press;2002.
Response: Thanks for the information regarding the sample size. As there are many ways to calculate the sample size, we have used Raosoft online calculator to calculate the sample size which is based on the total target population. The book cited by the reviewer is for the development of a questionnaire followed by exploratory and confirmatory factor analysis. This was not the objective of our study. Also, all the studies cited by the reviewer earlier have been done based on the study population and none of the studies has done exploratory and confirmatory factor analysis.
Comment: Authors’ response to Q. 8 is not convincing enough. How authors determined if their items/scale was unidimensional measuring emergency preparedness and perceived response? On the face value, the items appears to be assessing multi dimensionality? Therefore one-dimensionality of the questionnaire is still remains in question?
Response: Our previous response is as follows. Dichotomous response scale was used in this study as they are good for factual reporting as there are only two possible answers to analyze and report on. It’s either one thing or the other. They provide a clear distinction of qualities, experiences, or respondents’ opinions. They are also short and simple to analyse the data findings. They simplify the survey experience for the respondents to complete. However, there are some limitations to the dichotomous response scale. We have added the following in the limitations section. “As this study used dichotomous questions, the respondents did not get the opportunity to share their opinions”.
Moreover, the studies cited by the reviewer “Pandemic preparedness of community pharmacies for COVID-19” and “Community pharmacists' preparedness and responses to COVID‐19 pandemic: A multinational study” have also used yes/no questions only. As such, our study does not deviate from the other studies.
Comment: Authors’ response to Q. 11 – Authors should include this info into the relevant section of the manuscript.
Response: Thanks for the suggestion we have included the following under the heading validity and reliability. One of them is a Malaysian community pharmacist with 25 years of experience. Another expert is having 20 years of experience in social pharmacy and behavioural sciences research. Another expert is an associate professor in pharmacy practice with 18 years of experience.
Comment: Authors’ response to Q. 12 is again inappropriate. The reference provided here came from an answer provided to the question asked on ResearchGate. Moreover, the referred answer came from a deleted profile. What is the authenticity of such reference and what about its accuracy? Is this reference even professionally acceptable?
Here is some more information on when to use Spearman’s Rho analysis.
Comparison of the Pearson and Spearman correlation methods
Available at: https://support.minitab.com/en-us/minitab-express/1/help-and-how-to/modeling-statistics/regression/supporting-topics/basics/a-comparison-of-the-pearson-and-spearman-correlation-methods/
Spearman's Rank-Order Correlation. Available at: https://statistics.laerd.com/statistical-guides/spearmans-rank-order-correlation-statistical-guide.php
Use of inappropriate statistical test for data analyses, further raises the question about the reliability and validity of the results presented by the authors for their correlation analyses which is a crux of this study. Even if this part is excluded from the manuscript, the current study will remain simply a descriptive study offering no benefits to its readers or add anything to the published literature.
Response: Thanks for the detailed explanation. From the above references, we understand that rank correlation (non-parametric) needs to be done for correlation between a continuous and categorical variable. As such, we have used the rank biserial correlation to find the relationship between gender, ethnicity, type of pharmacy, state, (categorical variables), and PR scores (interval/ratio). However, like Spearman's rank, the rank biserial correlation also ranges from -1 to +1. As such, we did not get any variation in our results. Only the numerical values have changed. We have revised this in table 4 as well as in our manuscript. We have used the Spearman's Rank to find the relationship between age, experience in years, Number of pharmacists in a pharmacy, trained on disease outbreak management, Number of COVID-19 cases attended (ranked variables), and PR scores (interval/ratio).
Comment: Authors’ response to Q. 14 is partially true. The strength of the correlations and cut-off points are arbitrary and inconsistent (different source proposed different cut-off points) and should be used judiciously. Here are few references to expand on this----
Hinkle DE, Wiersma W, Jurs SG. Applied Statistics for the Behavioral Sciences. 5th ed. Boston: Houghton Mifflin; 2003.
Schober, Patrick MD, PhD, MMedStat; Boer, Christa PhD, MSc; Schwarte, Lothar A. MD, PhD, MBA Correlation Coefficients: Appropriate Use and Interpretation, Anesthesia & Analgesia: May 2018. Volume 126 - Issue 5 - p 1763-1768 doi: 10.1213/ANE.0000000000002864.
Response: Thanks for the references. After going through these references, we understand that our earlier response to the reviewer is acceptable. The response is as follows.
Perfect: If the value is near ± 1, then it is said to be a perfect correlation: as one variable increases, the other variable tends to also increase (if positive) or decrease (if negative).
High degree: If the coefficient value lies between ± 0.50 and ± 1, then it is said to be a strong correlation.
Moderate degree: If the value lies between ± 0.30 and ± 0.49, then it is said to be a medium correlation.
Low degree: When the value lies below + .29, then it is said to be a small correlation.
No correlation: When the value is zero.
Comment: Authors’ response to Q. 16 is also partially correct. Since, the self-cited article authors referring to showed significant differences between different healthcare professionals (GPs, RPh, Nurses, dentist etc.) about the emergency preparedness and perceived response for Ebola outbreak. Pharmacists participated in this study had low mean score on both scale compared to general practioners and nurses. This again, doesn't adequately justifies the rationale for the use of 50% cut-off score.
Response: Yes, the reference provided here is based on the authors’ previous publications. The previous studies were on health professionals and that included community pharmacists too. There is no drastic variation in the study population as all of them are healthcare professionals. Hence, we used the 50% cut-off point for this study as well.